# Glycosphingolipids in Osteoarthritis and Cartilage-Regeneration Therapy: Mechanisms and Therapeutic Prospects Based on a Narrative Review of the Literature

**DOI:** 10.3390/ijms25094890

**Published:** 2024-04-30

**Authors:** Kentaro Homan, Tomohiro Onodera, Masatake Matsuoka, Norimasa Iwasaki

**Affiliations:** Department of Orthopaedic Surgery, Faculty of Medicine and Graduate School of Medicine, Hokkaido University, Kita 15, Nishi 7, Kita-ku, Sapporo 060-8638, Japan; k.houman@med.hokudai.ac.jp (K.H.); masatakem@pop.med.hokudai.ac.jp (M.M.); niwasaki@med.hokudai.ac.jp (N.I.)

**Keywords:** glycosphingolipids (GSLs), osteoarthritis, articular cartilage, gangliosides, chondrocyte differentiation, cartilage regeneration

## Abstract

Glycosphingolipids (GSLs), a subtype of glycolipids containing sphingosine, are critical components of vertebrate plasma membranes, playing a pivotal role in cellular signaling and interactions. In human articular cartilage in osteoarthritis (OA), GSL expression is known notably to decrease. This review focuses on the roles of gangliosides, a specific type of GSL, in cartilage degeneration and regeneration, emphasizing their regulatory function in signal transduction. The expression of gangliosides, whether endogenous or augmented exogenously, is regulated at the enzymatic level, targeting specific glycosyltransferases. This regulation has significant implications for the composition of cell-surface gangliosides and their impact on signal transduction in chondrocytes and progenitor cells. Different levels of ganglioside expression can influence signaling pathways in various ways, potentially affecting cell properties, including malignancy. Moreover, gene manipulations against gangliosides have been shown to regulate cartilage metabolisms and chondrocyte differentiation in vivo and in vitro. This review highlights the potential of targeting gangliosides in the development of therapeutic strategies for osteoarthritis and cartilage injury and addresses promising directions for future research and treatment.

## 1. Introduction

Osteoarthritis (OA), a prevalent joint disorder, imposes a significant economic strain, costing the medical economy over USD 80 billion annually [1,2,3]. Despite extensive research, the pathogenesis of OA, marked by the progressive deterioration of articular cartilage and extracellular matrix (ECM), remains largely elusive [4]. The articular cartilage, also known as hyaline cartilage, is required by its elasticity, crucial for load absorption and distribution, and its smooth, lubricated surface, which facilitates motion and minimizes friction [5]. The avascular, aneural, lymphatic, and hypocellular nature of articular cartilage, a specialized connective tissue, restricts its access to nutrients and circulating chondrogenic progenitor cells, thereby limiting its intrinsic healing capacity [6,7]. This limitation can promote cartilage degeneration, culminating in OA. Recent attention has been drawn to the potential role of glycolipids in OA pathogenesis, following the discovery of significant alterations in the composition of glycosphingolipids (GSLs) in the articular cartilage of OA patients [8,9,10]. This has led to the consideration of biomembrane glycolipids as potential contributors to OA pathogenesis in post-genomic studies.

GSLs are key components of cell membranes, comprising a hydrophobic ceramide and a hydrophilic oligosaccharide residue (Figure 1). Ceramides are embedded in the outer leaflet of the plasma membrane, while oligosaccharides project into the extracellular space [11,12]. GSLs cluster on the cell-membrane surface, modulating transmembrane signaling and mediating intercellular and cell–matrix interactions [11,12,13,14]. An enzyme called glucosylceramide synthase encoded by the Uridine diphosphate (UDP)-glucose ceramide glucosyltransferase (Ugcg) gene is responsible for directing the first committed step in GSL synthesis [11,15,16,17]. Glucosylceramide is formed when a glucose moiety is transferred from UDP-glucose to ceramide, which is the precursor of most cellular GSLs. Mice with a global disruption in UGCG are embryonically lethal (E7.5), suggesting that GSLs are essential for embryonic development and differentiation [15,16,17,18]. It is now well established that some sphingolipids can regulate key biological functions, and these include cell growth and survival, cell differentiation, angiogenesis, autophagy, cell migration, and organogenesis [19]. GSLs are expressed not only in cartilage, but also in the nucleus pulposus tissue of the intervertebral disc [20], and are also abundant in nerve tissue [15,21,22], suggesting an association with pain [23,24]. Furthermore, specific bioactive sphingolipids have been linked to various pathologies, including inflammation-related diseases like atherosclerosis, rheumatoid arthritis, type II diabetes, obesity, cancer, and OA.

The role of GSLs in articular cartilage, OA pathogenesis, and therapeutic prospects has not provided a comprehensive overview of the role of GSLs in articular cartilage. A brief description of the role of GSL on cartilage, focusing on the processes of homeostasis and the differentiation of chondrocytes, will be followed by an explanation of the endogenous ability of articular cartilage to heal. Finally, the usefulness and prospects of GSLs expressed on cell membranes as biomarkers for quality control in cartilage-regenerative medicine and as therapeutic target molecules for OA will be discussed.

## 2. Impact of GSLs on Cartilage Homeostasis

Ceramide, a principal constituent of glycolipids, instigates the mRNA expression of collagenase-1/matrix metalloproteinase (MMP)-1 and stromelysin-1/MMP-3 in human fibroblasts via the activation of three distinct mitogen-activated protein kinases (MAPKs), extracellular signal-regulated kinase (ERK)1/2, stress-activated protein kinase/Jun N-terminal-kinase (SAPK/JNK), and p38 in cartilage [25]. Moreover, ceramide has also been discovered to be implicated in cartilage degeneration and apoptosis [26]. The ceramide pathway activator curtailed the production of inflammatory cytokines (interleukin [IL]-1β, IL-6, and IL-18) and the activation of the MAPK pathways (p-ERK, p-JUK, and p-p38) [27]. As mentioned above, systemic knockout mice of the *UDP-glucose ceramide glucosyltransferase* gene prove to be embryonically lethal because UGCG is the initial committed step in the synthesis of the majority of GSLs [11,15,16,17]. GSLs form clusters on the plasma membrane and undertake diverse roles in regulating membrane-mediated signal transduction and in mediating cell–cell and cell–extracellular matrix interactions [28,29,30,31]. Consequently, an endeavor is undertaken to identify even the most characteristic downstream glycolipid molecules by the sequential knockout of upstream glycosyltransferase genes implicated in the impairment of cartilage homeostasis in chondrocytes. The contents of such research studies are encapsulated in Figure 2 and Table 1.

A decrease in all major gangliosides, contrasting with a marked increase in the monosialodihexosylganglioside (GM3), has been demonstrated in osteoarthritic fibrillated cartilage. Some previous studies have shown that gangliosides have tissue-protective effects against oxidative stress or apoptosis in neuronal, cardiac, and hepatic cells [32,33,34,35,36]. The results of the series of studies indicate that the loss of gangliosides results in greater cartilage vulnerability to IL-1α/β stimulation in the cartilage-degradation process by increasing MMP-13 secretion and chondrocyte apoptosis. The mechanical properties of cartilage have been mainly focused on hyaluronan and chondroitin sulfate [37,38]. Since these increases and decreases affect the physical properties of cartilage, they have been relatively well studied and are attracting attention as a therapeutic approach [39]. Sphingomyelin is involved in the boundary lubrication of articular cartilage [40], and the presence of GSLs on chondrocytes appears to raise the threshold of sensitivity to mechanical stress and resist catabolic reactions [41]. This is not unrelated to the fact that GSLs are expressed on the plasma membrane. On the other hand, there are indications that replenishing the cells with the missing gangliosides can restore normal activation [42,43]. The fact that β1, 4-N-acetylgalactosaminyltransferase (GalNAcT) knockout mice must be supplemented with all three series to be rescued indicates that each series of gangliosides is essential for cartilage maintenance (Figure 2) [43]. Considering previous studies [18,42,43,44], a treatment supplementing the o-, a-, and b-series gangliosides below GalNAcT appears promising, with future therapeutic trials anticipated.

Glycosidase inhibitors are also considered to be an important target for cartilage regeneration. They are directly linked to osteoarthritis because N-acetyl-beta-hexosaminidase is the predominant glycosidase released by chondrocytes to degrade glycosaminoglycan [45]. The stimulation of chondrocytes with IL-1β selectively increases extracellular hexosaminidase activity among many enzymes, suggesting that hexosaminidase is the cartilage matrix-degrading enzyme activated by inflammatory stimuli [45]. The inhibitor of this hexosaminidase is shown to modulate intracellular levels of glycolipids, including Galβ1,4(Neu5Acα2,3)Galβ1,4Glc (GM2) and asialo ganglioside GM2 (GA2) (o- and a-series gangliosides) [46]. Since these kinetics have been studied in adults and in the elderly who have formed permanent articular cartilage, caution should be exercised when applying them to children or infants.

**Table 1 ijms-25-04890-t001:** Genetic defects in mouse glycan formation and physiologic consequences.

Glycosyltransferase	Lost Glycolipids	Consequences of Depletion of Its Glycolipid	References
UGCG (Glucosylceramide synthase)	GSLs	Embryonic death. Reduced insulative capacity of the myelin sheath. Col2-*Ugcg*^−/−^ mice enhance the development of OA.	[16,17,18,41,47,48]
ST3GalIV (GM3S)	Gangliosides other than the o-series	GM3 plays an immunologic role. Heightened sensitivity to insulin. Severely reduced CD4+ T cell proliferative response and cytokine production. Promotes OA and RA but cartilage regeneration.	[42,44,49,50,51]
ST8SiaI (GD3S)	b-series ganglioside	Tumor-associated carbohydrate antigens (TACA) in neuro-ectoderm-derived cancers. Suppression of age-related bone loss. Deteriorates OA with aging.	[43,52,53,54,55]
GalNAcT (GM2/GD2S)	Almost all gangliosides except GM3, GD3, and GT3	Age-dependent neurodegeneration and movement disorders associated with it. Defects in spermatogenesis and learning. Exacerbating OA progression.	[43,56,57,58]

## 3. Role of GSLs in Cartilage Repair and Differentiation Processes

GSLs play a crucial role in the repair and differentiation processes of articular cartilage [50,59]. These processes are essential for addressing cartilage defects, commonly observed in osteoarthritis. The unique properties of GSLs facilitate the regeneration of cartilage tissue, which is key to restoring joint function [43,60]. Understanding the mechanisms behind cartilage regeneration, particularly the role of GSLs, is critical.

Here, various aspects of cartilage repair will be explored, with a focus on the role of glycolipids in promoting the regeneration and repair process through chondrogenic differentiation.

### 3.1. Endogenous Potential to Heal in Articular Cartilage

To enhance endogenous cell recruitment to the injury site, the biological process within a living organism after articular cartilage injury needs to be clarified. Models have been constructed in mice [61,62,63], rats [64], rabbits [65,66,67], horses [68], and canines [69] to study the repair process of articular cartilage. This has successively revealed various molecular reactions after articular cartilage injury [70,71,72,73,74]. Even though the species are different, the major ganglioside in articular cartilage appears to be GM3 [75,76]. GSLs consist of several types of glycolipids and are classified into several groups depending on their structural features, which include neo-lacto-series, globo-series, isoglobo-series, and ganglio-series (gangliosides) [28,44]. GM3 functions as a precursor molecule for the majority of the more intricate ganglioside species [49], and ganglioside expression patterns in cells during differentiation undergo alterations in response to cytokine and growth factor stimulation [77,78,79]. Many receptor tyrosine kinases, including the epidermal growth factor receptor (EGFR), an N-glycosylated transmembrane protein with an intracellular kinase domain, are localized in lipid rafts and inhibit the tyrosine kinase activity of EGFR through the inter-glycan interaction between N-glycans on EGFR and oligosaccharides on GM3 [78,79]. Equally, GM3 inhibits vascular endothelial growth factor (VEGF)-induced activation of VEGF receptor-2 (VEGFR-2) by blocking VEGF dimerization and inhibits VEGF binding to VEGFR-2 through GM3-specific interactions with the extracellular domain of VEGFR-2 [77]. Predicated on the outcomes of these in vitro studies, it is anticipated that GM3 would assume a significant role in the process of articular cartilage repair, as it is implicated in the regulation of tyrosine phosphorylation of growth factor receptors and angiogenesis. During the articular cartilage healing process, GM3 has dual roles in recruiting chondrogenic precursor cells to the injury site and in the induction of hypertrophic differentiation in chondrocytes [50]. The manipulation of ganglioside expression may be the future direction in articular cartilage regeneration. Accordingly, a site- and time-specific intervention is needed to manipulate glycosphingolipids in articular cartilage. Genome editing can overcome these difficulties, with CRISPR (clustered regularly interspaced short palindromic repeats)-Cas9 being the simplest and most rapid technology [80]. Previously, CRISPR-Cas9 has been used to modify GSL glycans on leukocyte cells [81], and it might be useful for regenerating cartilage [82]. The multi-omics of cartilage from a spatio-temporal perspective is currently of vital interest [83,84]. On the other hand, spatial and temporal analysis of glycome expression patterns on tissues is limited to N-glycans due to technical difficulties [85,86,87,88]. Advances in innovative spatial biology techniques are expected to allow intact tissue sections to be examined using the glycome of GSLs with spatial data.

### 3.2. Changes in the Glycan Structure during Chondrogenic Differentiation

Chondrogenic differentiation is a well-organized process; cartilage is formed from condensed mesenchymal tissue that differentiates into chondrocytes and begins to secrete the molecules that make up the extracellular matrix [89]. Extracellular enzymes, which include the matrix metallopeptidases, lead to the activation of cell signaling pathways and gene expression in a temporal-spatial-specific manner during the development process. The recruited mesenchymal stem cells may attempt to differentiate into chondrocytes after articular cartilage injury [90,91]. However, the response after injury is not a fully recapitulated process of development, resulting in regenerated cartilage-like tissue that does not possess typical biomolecules of hyaline cartilage such as type II collagen and aggrecan, and a proportion of their chemical constitutes differ from those in the original cartilage [92,93]. Molecules promoting the selective differentiation of multipotent mesenchymal stem cells into chondrocytes have been reported to stimulate the repair of injured articular cartilage (MMP14, microphthalmia-associated transcription factor [MTIF], Glycoprotein Nmb [GPNM], Secreted Phosphoprotein 1 [SPP1], Prostaglandin-Endoperoxide Synthase 2 [PTGS2], Vascular Endothelial Growth Factor A [VEGFa], and JunD Proto-Oncogene [JUND] [94]. Therefore, the regulation system for chondrogenic differentiation is attracting attention.

Glycosylation is one of the post-translational modifications in cell-surface proteins and extracellular matrix proteins which regulate a variety of biological functions, including the enhancement of protein stability, controlling cell-to-cell communication, and adhesion [95]. In addition, this process is known to contribute to the pathogenesis of various kinds of diseases [96,97]. Quantitative and qualitative analyses of N-glycans during chondrogenic differentiation were previously performed using the glycoblotting method [98], followed by glycoform-focused reverse proteomics and genomics using a mouse pre-chondrogenic cell line, an embryonal carcinoma-derived chondrogenic cell line ATDC5 [99]. The levels of high-mannose-type N-glycans increase during chondrogenic differentiation, suggesting that N-glycans may have key roles in the differentiation and/or homeostatic maintenance of chondrocytes [100]. As for hypertrophic differentiation in chondrocytes, Yan et al. reported that resting chondrocytes exposed to concanavalin A, which binds specifically to high-mannose-type structures, selectively differentiated to the hypertrophic stage [101,102]. Concanavalin A is known to be a potent mitogen and is found to promote differentiation by inducing the cross-linking of high-mannose-type N-glycans. A comprehensive analysis of all classes of glycoconjugates on articular cartilage, including N-glycans, O-glycans, free oligosaccharides (fOSs), glycosaminoglycans (GAGs), and GSLs, revealed dynamic alterations [76,103]; the quantitative glycan profile showed that several N-glycans increased significantly with hypertrophy, whereas GSL and fOS decreased significantly. These findings suggest that glycan markers can be used as differentiation biomarkers for chondrogenic differentiation and may help to evaluate the regenerative product after articular cartilage injury.

## 4. Cell Sources

As stated in the Introduction Section, articular cartilage is a type of hyaline cartilage that enables smooth movements between bones in articulating joints, which requires both weight-bearing and low-friction capability [104]. Therefore, cartilage regeneration with these high qualities is important. The ultimate goal for ideal cartilage regeneration is to restore these key properties of the original hyaline cartilage in terms of histological structure and biomechanical functions, which seems to be only achieved by replacing it with healthy cartilage tissue [105]. Several types of cell-based approaches have been introduced at the present moment [106]. Representative strategies include autologous cartilage implantation, mesenchymal stem cells, and induced pluripotent stem cells. Table 2 summarizes three representative cell types and reports on cartilage regenerative medicine. This article covers cell-based regenerative medicine in articular cartilage, primarily based on glycobiology.

### 4.1. Autologous Chondrocyte Implantation

Autologous chondrocyte implantation is the beginning of a major trend in regenerative medicine for cartilage injuries, published by Lars Peterson and his group in Sweden in 1994 [144]. Articular cartilage does not possess access to the nutrients or circulating chondrogenic progenitor cells, and cartilage lacks the natural potential to overcome a sufficient healing response by possessing a nearly acellular nature [145]. Consequently, articular cartilage has limited healing potential; therefore, it can lead to cartilage degeneration and ultimately result in OA. Autologous chondrocyte implantation is conceived to compensate for the sparse and nonvascular nature of cartilage by providing cultured chondrocytes: a cell-based therapy consisting of two-staged procedures for full-thickness defects of articular cartilage in the knee [144]. This procedure requires the first harvest of chondrocytes from a non-weight-bearing area of articular cartilage. After a culture of 4 to 6 weeks, a second-stage procedure is undertaken to implant amplified chondrocytes into the defect. Considering the limited healing potential for articular cartilage, these procedures seem to be ideal. 

Since the first clinical report was published, several authors have demonstrated successful clinical outcomes of this procedure for cartilaginous lesions [146,147,148]. However, there remain concerns about the dedifferentiation of chondrocytes during the culture period due to the limited proliferative capacity [149,150,151]. The cartilage extracellular matrix possesses various glycosylated proteins, which contribute to the maintenance of its specific functions [152]. Sialic acids are negatively charged sugars expressed at the terminal positions of N- and O-linked oligosaccharides, which are attached to cell surfaces or secreted glycoproteins. As a result of their non-reducing terminal position, sialic acids are involved in highly specific recognition phenomena [153,154]. Primary human chondrocytes predominantly express α2,6-specific sialyltransferases and α2,6-linked sialic acid residues in glycoprotein N-glycans [155]. Interestingly, inflammation stimuli induced a shift from α2,6-linked to α2,3-linked sialic acid, suggesting that α2,6-linked sialic acid can be used as a biomarker for quality control in amplified human cartilage.

### 4.2. Mesenchymal Stem Cells (MSCs)

Mesenchymal stem cells are multipotent stem cells and can be obtained from various organs including bone marrow, synovium, periosteum, adipose tissue, and skeletal muscle [156]. MSCs are attractive cell sources in regenerative medicine based on their abilities to self-renew and differentiate into mesenchymal tissue lineage [157]. During the last few years, the use of MSCs and their cell-free derivatives has seen an increasing number of applications in disparate medical fields, including chronic musculoskeletal conditions [60,158,159,160,161]. The advantage of autologous MSC transplantation is that it avoids the problem of defects caused by cartilage harvesting to secure cells in autologous cartilage transplantation. MSCs are not considered to be tumorigenic like ES cells, and only minor side effects such as fever, chills, and liver damage have been reported as side effects of MSC therapy. Furthermore, the potency of MSCs can be enhanced using the bone marrow aspirate concentrate (BMAC) method [142,162]. BMAC therapy has a slow onset of benefit (2 to 4 years) [143], so cell quality is crucial. One of the concerns is that MSCs are heterogeneous populations, whose capability to differentiate varies depending on the tissues harvested and the donor age. MSCs are grouped by the common characteristics of CD44, 73, 90, and 105 positivity and CD31 and 45 negativity, but these do not necessarily define a “stem cell”. In addition, since the properties of these cells change depending on the isolation method, culture conditions, and passages when they are grown in vitro until the required cell number is reached, there has been a need for an appropriate biomarker that could define “stem cells”. Considering how OA cartilage is characterized by a reduction of most gangliosides, these gangliosides could be combined with expression information such as high-mannose-type N-glycans, linear poly-N-acetyllactosamine chains, and α2,3-sialylation to more accurately define MSC undifferentiated characteristics.

Glycan expression changes rapidly upon differentiation; thus, many of the typical cell markers are glycans [163,164]. This is in contrast to the little change in the protein expression profiles between differentiated and undifferentiated cells [165]. Glycosylation features associated with bone marrow-derived MSCs included high-mannose-type N-glycans, linear poly-N-acetyllactosamine chains, and α2,3-sialylation [166]. Their cellular differentiation stage can be determined using these glycomics. Tateno et al. carried out glycome analysis on different passages of adipose-derived human MSCs (hMSCs) using high-density lectin microarrays to identify glycan markers that distinguish MSCs to have enough capability to differentiate [167]. This report indicated that α2,6-linked sialic acid-specific lectins showed stronger binding to the early passage of adipose-derived hMSCs with the ability to differentiate to adipocytes and osteoblasts than late passage cells without the ability did. They also reported quantitative glycome analysis targeting both N- and O-glycans from early and late passages of adipose tissue-derived hMSCs and showed that the expression of α2,6-sialylated N-glycans varies depending on the differentiation potential of stem cells but not O-glycans, suggesting that α2,6-sialylated N-glycans can be used as biomarkers for the quality control of hMSCs [168]. The presence of α2,6-linked sialic acid structure is a characteristic of pluripotent stem cells that possess higher differentiation potential. This may serve as an indicator of their differentiation potential. Ryu et al. showed that the gangliosides GM3 and GD3, which contain α2,3- and α2,8-linked sialic acids, were expressed after the chondrogenic differentiation of synovium-derived hMSC aggregates [59]. In the same way, GM3 expression increased temporarily following the chondrogenic differentiation of hMSCs derived from bone marrow [169]. Considering OA cartilage is characterized by a decrease in most gangliosides, these gangliosides may be useful in developing therapeutic agents for MSC-based articular cartilage regeneration in articular cartilage disease. 

### 4.3. Induced Pluripotent Stem Cells (iPSCs)

In 2006, Takahashi and Yamanaka reported pluripotential stem cells from mouse embryonic or adult fibroblasts by introducing four factors, Oct3/4, Sox2, c-Myc, and Klf4 [170]. The method to establish human iPSCs dramatically evolved and simplified [171]. This progress provides us with opportunities to understand the disease mechanisms and promote regenerative medicine [172,173]. To avoid the rejection of differentiated cells originating from iPSC transplantation, autogenous transplantation with iPSCs originating from the individual’s own cells is ideal. However, these procedures require time to establish iPSCs of high enough quality for transplantation as well as the cost. In contrast, iPSCs induced from healthy donors with a homozygous human leukocyte antigen haplotype (HLA-homo) is a significant candidate for allogenic transplantation on the basis that HLA-homo iPSCs might not be rejected by HLA haplotype-matched patients [174,175,176]. iPSC banking uses cells recruited from healthy, consenting HLA-type homozygous donors and is made with peripheral blood-derived mononuclear cells or umbilical cord blood [177]. Many research groups have been trying to apply iPSCs-based therapy to patients, and some of them are already being administered in clinical trials [178]. The allogeneic transplantation of iPS cell-derived cartilage has shown comparable results to the allogeneic transplantation of polydactyly-derived chondrocyte sheets in preclinical studies [129], indicating its potential to provide a vast cellular resource for creating artificial cartilage tissue [128,130]. On the other hand, the argument remains that there are differences between iPSCs and their derivatives from healthy and diseased donors [131], and proving the quality of iPS cell-derived cartilage is key to optimization.

As for cartilage metabolisms, the main techniques to confirm chondrogenic differentiation from iPSCs are based on the detection of upregulated chondrogenic genes or the histological analysis of the extracellular matrix. When human iPSCs, iPSC-derived MSC-like cells, iPS-MSC-derived chondrocytes (iPS-MSC-CDs), and bone marrow-derived MSCs are induced to differentiate into chondrocytes, respectively, the cartilage ultimately produced is comparable in general evaluation (type 2 collagen, aggrecan) but different in GSL-glycan profile [169,179]. It remains uncertain how this difference in glycan structure will behave in vivo. These data should, however, assist in assessing the quality of cartilage obtained through modern regenerative medicine.

In terms of tumorigenicity for iPSCs themselves, while clinical applications are going forward, the concerns that the transplantation of differentiated iPSC might lead to teratoma formation in the recipient should be clarified [180,181]. Matsumoto et al. reported that the R-17F antibody detects undifferentiated iPSCs harboring the Lacto-N-fucopentaose I (LNFP I) of GSLs, and as a result, exerts cytotoxic activity [182]. Recently, several methods have been developed to identify sialic acid-linked isomers on glycans [183,184,185], and α2,3-linked sialic acids on glycolipid glycans can also be detected [186]. If teratoma formation can be prevented by modifying the glycan structures involved in maintaining the undifferentiated nature of iPS cells, iPS cells can be used more safely [187,188]. The GSLs–glycome analysis is useful to determine the optimal conditions for removing undifferentiated iPSCs to a level safe for transplantation.

## 5. Conclusions

The field of GSLs and their biological functions is still in its infancy and lacks a rigorous review process such as defined inclusion and exclusion criteria. This limitation may affect the comprehensiveness and accuracy of the review. Despite these challenges, the impact of glycolipids on cartilage is not negligible, and a glycolipid-based cartilage-regeneration strategy is attractive. Although there have been no clinical trials using GSLs in cartilage-treatment strategies, and all of the related literature is in the experimental phase, the glycomics of mesenchymal stem/progenitor cells can be utilized to evaluate their stage of cellular differentiation. Furthermore, this review highlighted the possibility that supplementing missing GSLs could play significant roles in tissue regeneration and disease modification. To guide the regeneration of degenerated or injured cartilage into articular cartilage, a multifactorial methodology that incorporates GSLs, which are closely related to cartilage homeostasis, should be developed in the future.

## Figures and Tables

**Figure 1 ijms-25-04890-f001:**
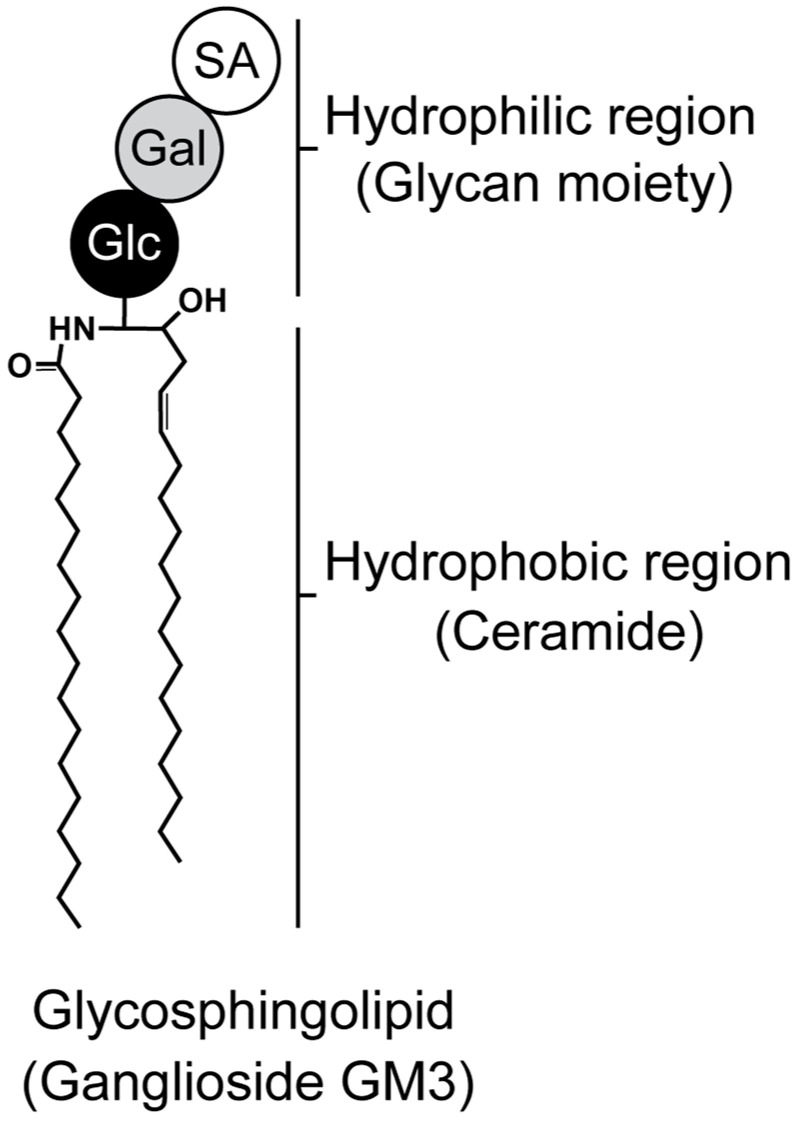
Structure of glycosphingolipid. Ganglioside monosialodihexosylganglioside (GM3) is composed of both hydrophilic (glycan) and hydrophobic (ceramide) regions. Glc, Gal, and SA indicate glucose, galactose, and sialic acid, respectively.

**Figure 2 ijms-25-04890-f002:**
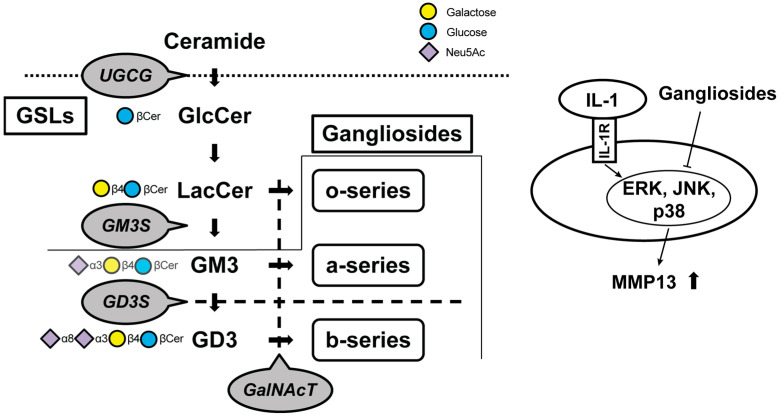
Schematic of the biosynthetic pathway for gangliosides. Glucosylceramide (GlcCer) synthase, encoded by the *UDP-glucose ceramide glucosyltransferase (Ugcg)* gene, synthesizes GlcCer from ceramide. Gangliosides are classified as o-, a-, or b-series according to the number of sialic acids attached to galactose. Monosialodihexosylganglioside (GM3) synthase (GM3S) is required for the GSL synthesis downstream of lactosylceramide (LacCer), including the a-series and b-series. The b-series gangliosides are synthesized from the common precursor molecule Disialosyllactosylceramide (GD3), which is the product of GD3 synthase (GD3S, encoded by the *Gd3s* gene). β1, 4-N-acetylgalactosaminyltransferase (GalNAcT) activity is required for the elaboration of the o-, a-, and b-series precursors LacCer, GM3, and GD3, respectively. Gangliosides suppress protease upregulation through the mitogen-activated protein kinase signaling pathways. Cer, ceramide; GSLs, glycosphingolipids; IL, interleukin; MMP, matrix metalloproteinase; ERK, extracellular signal-regulated kinase; JNK, c-Jun N-terminal kinase; p38, Thr180/Tyr182.

**Table 2 ijms-25-04890-t002:** Cell sources and cartilage-regenerative medicine.

Clinical Practice	Cell Source	Lesion Size (cm^2^)/OA Grade	Performances	References
Microfracture	Mesenchymal stem cell (MSC)	2.0–4.0	Microfracture is most likely to be successful for small femoral condylar defects.	[107,108,109,110,111]
Autologous matrix-induced chondrogenesis (AMIC)	MSC	1.3–5.3	Effective procedure for the treatment of mid-sized cartilage defects. Low failure rate with satisfactory clinical outcomes.	[107,108,112,113,114,115,116,117,118]
Autologous chondrocyte implantation	Chondrocyte	2.0–10.0	Superior structural integration with native cartilage tissue compared to microfracture and AMIC, but a two-stage treatment burden exists.	[108,119,120,121,122]
Osteochondral autograft transplantation	Chondrocyte	0.1–20.0/OA grade I–III	Osteochondral autograft transfer system and mosaicplasty appear to be an alternative for the treatment of medium-sized focal chondral and osteochondral defects of the weight-bearing surfaces of the knee. Chondrocyte sheet and auricular cartilage micrograft for the treatment of early-stage OA has been tried.	[123,124,125,126]
Allogenic transplantation	Chondrocyte, iPSC	2.2–4.4/OA grade II–IV	Osteoarticular allograft transplantation is used to treat high-grade cartilage defects or arthritis. iPSC-derived cartilages are used in preclinical studies that are in the middle to late stages when clinical trials are within range.	[127,128,129,130,131,132,133,134]
Intra-articular injection with stem cell	Adipose-derived stem cell, MSC	OA grade II–IV	Lower degenerative grades improve outcomes but are less effective for end-stage OA. The results of intra-articular administrations of stem cells are better with BMSC. In particular, the use of bone marrow aspirate concentrate (BMAC) is also indicated for severe OA.	[135,136,137,138,139,140,141,142,143]

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
