# Peer review of "Glycosphingolipids in Osteoarthritis and Cartilage-Regeneration Therapy: Mechanisms and Therapeutic Prospects Based on a Narrative Review of the Literature"

_ijms, 2024, doi:10.3390/ijms25094890_

Round 1
Reviewer 1 Report
Comments and Suggestions for Authors
I would to congratulate the authors for putting such effort into an interesting and well-tailored review.
All things considered, it is my duty to help you reach an even higher tier of excellence. Therefore all my corrections are in the pursuit of this goal. Please, apply them for a successful submission.
Acronyms: Please define all acronyms the first time that they appear in the main text. This is mandatory. Ex: GM3, GM2…etc.. Also, authors should be careful when defining ex: IL-1 as not the same as IL1β. As such, the statement and reference should be revised. The whole manuscript requires a thoughtful revision in this matter. Moreover, figure legends should be stand-alone and readable. Acronyms should be defined here always. Some exceptions might be made here to avoid oversized legends.
References: All of the statements require a reference. All. Any statement without reference is an opinion of the author and should be written with the proper wording.
· Ex: The inhibitor of this hexosaminidase was shown to modulate intracellular levels of glycolipids, including GM2 and GA2 (o- and a-series gangliosides).
· Similarly, in Lines 120-124; 144-16 the list is very long. Revise the whole manuscript.
· Self-citation: An overrepresentation of self-citations was detected by all authors. Please be very careful in this to avoid manuscript rejection.
Clinical, in vivo & in vitro fact: To reach conclusions from different disease models you may separate data among different clinical, mice & cell models having each a different paragraph. If not, every statement must reflect the cell model. For example: Lines 140 -144 introduce some chronological events out of the blue from experiments performed in mice (reference 42). Please improve what you are trying to say.
The manuscript lacks inclusion and exclusion criteria for the manuscripts selected for the review. This is necessary to avoid biased reviews.
The whole manuscript focuses on the GLS and its importance but does not properly explain the dynamics with surrounding tissues and their importance physiologically and as therapeutic targets.
Authors use references in OA as well as cartilage regeneration at the physiological or therapeutic level such as Lines 105-112. However, these dynamics change dramatically in newborns, children, adults, and/or elders. Please revise these statements whether are applied to all ages or a particular population.
Limitations, efficiency, after effects section: Across the manuscript, there seems to be a subjacent message that GLS is meant to be golden standards. A similar situation happens to the different techniques depicted in section 4.1 Please clearly state the advantages and limitations of each technique or the use of GLS as a biomarker or therapeutic target. An example is in Lines 219-223. Similarly, it is difficult to elucidate from the manuscript whether each technique in section 4.1 is currently being clinically approved considering that a quarter of a century has passed. We can’t elucidate potential synergies or interferences and /or when it is recommended a technique to be used. More importantly, we have no references or reports showing any effect or adverse effects. Limitations should be addressed in each subsection or a full paragraph.
Other sections to revise:
· Section 4.1.: Names lack dots. The first paragraph should be rewritten to improve text comprehension.
· Subsection 4.2: This section is not well explained. I would advise rewriting it from top to down and extending the text as necessary. As mentioned before, the authors do not properly address
· Lines 274-284: The authors introduce a hot topic from two decades ago. iPSCs. The authors do not properly explain the challenges that this technology faces including from HLA-homo. There are some wording problems regarding the use of “donors” and “individuals”.
· Lines 301: Please restrain the use of “we” across the manuscript. In this particular case, it is unknown whether all authors participated in the SALSA method as well as if there are some authors not credited. You may use X group or X author et al. Plus do not avoid mentioning alternatives to SALSA.
· A find amiss current technologies aligned to the review such as CRISPR Cas9 and Spatial OMICs. Any thoughts on that?
· Conclusion: Please be more specific in the stage that GSLs are considering the literature you used to back up your review. Is it in the preclinical stage and/or experimental use?
Other minor problems include:
· A mixture of verb tenses. Check Lines 105-112. But it can be found across the manuscript.
· Capital letter: lines 100-101
· Figures: I found a figure reflecting section 4.
Comments on the Quality of English LanguageEnglish: Generally speaking there are no major mistakes, text cohesion is fine, plus writing style is kept across the manuscript. Nonetheless, several sentences need new wording. Those are not the proper words for multiple reasons (incorrect, imprecise…etc). Some examples:
· Line 101 “There have been indications that…”
· Line 103 “It is reasonable to consider a”
· Line 192 “requires these high qualities of properties”
· Line 195 “As for now”…
· Line 196 “Representative strategy includes..” that should be “Representative strategies include”
· Similarly: Lines 130-134; 148-150; 152-154…. etc. Revise the whole manuscript.
Reviewer 2 Report
Comments and Suggestions for Authors
I would like to thank the editor for permiting me overview the manuscript
From thaking into consideration authors previous work, they are experts with extensive knowledge.
just a few minor points
Line 77 would be nice to mention the specific cytokines
Line 125 the use of the Word chapter, twhile tis is a review it is not a chapter
I would recommend a depiction of GM3 a nice figure would be Good to have
Line 162, whioch molecules a quick example
Line 173 while ATDC5 is an excelente model, should state it is derive from a teratoma just for purposes of acknowledging cancerous background and potential for higher glycolisation
Line 177 what mechanisim is involved in concavalin A differentiation
Really nice overview on MSCs and IPSC´s
Just a thought, would it be worth it to mention trans differentiation or direct differentiation, you mention MSC, IPSCs, some maturation form embryo, but is there any other work that crosses over from an indirect cell line… this point is not a make or break, on the contrary is simply as a question, if it can be addressed would be great.
Comments on the Quality of English Language
For the most part looks good, just at the star of the second subsection
2. Impact of GSLs on the cartilage homeostasis
After it was shown that a major component of glycolipids (ceramide) stimulates the
feels a bit rough
Reviewer 3 Report
Comments and Suggestions for Authors
The abstract is very short and non-applicable. The following should be added to it.
· Innovation and purpose of research
· Quantitative results and finally achievements
The article needs general writing and grammar editing.
The first paragraph of the introduction presented is primarily general information. At the end of the introduction, a suitable summary of the importance of the present issue should be provided. Also, discontinuity between paragraphs is evident in most of the introduction. In addition, the introduction should be rewritten. Its volume is very large.
Reviewer 4 Report
Comments and Suggestions for Authors
Reference lines 108 to 112 should be mentioned.
Different cell sources such as bone marrow should be investigated.
Comments on the Quality of English Languageno comment
Reviewer 5 Report
Comments and Suggestions for Authors
Cartilage regeneration has been of immense interest to the research community and this review summarizes the need and importance of Glycosphingolipids pretty nicely. In my opinion, the review requires a little attention on one section which can be addressed in minor revision, before being considered for publication
Major:
Knee cartilage is a load bearing joint as it is once again mentioned by the authors in the review. Mechanical properties of such joint is of paramount importance. Functionally, the challenges of regenerated cartilage extend to their capabilities of handling the mechanical load in comparison to native cartilage. My suggestion to the authors is to include a section of cartilage mechanical properties and how glycoshingolipids could help in regulation of mechanical properties. In the recent past, there has been various studies that has been published with the focus on mechanobiology of the cartilage.
Minor:
Line 78 uscg gene. Please use proper gene nomenclature
Change embryonic to embryonically
Line 103: Based on previous studies: please references the said studies again. Last sentence does not flow well with the line in 102-103. Please consider integrating the last sentence “Future therapeutic trials are pending” with the previous sentence.
Consider revising Line 130-134 “Considering species differences…. Appears to be GM3“ Information given in these sentences are immense but the delivery structure can be reconsidered.
Line 148: Expression may-future direction. Missing “be the”? Connection word is missing
Comments on the Quality of English Language
Nil
Reviewer 6 Report
Comments and Suggestions for Authors
This paper is a review on focusing on utilising gangliosides in tissue engineering approaches for cartilage injury treatments and osteoarthritis in general. The review follows a holistic approach and summarises the current available bibliography. It is well written and the different aspects are well explained.
Minor comment
189: “As we explained in the Introduction section” please change to: as stated in the introduction section
1. Originality and Relevance for the Field and specific gap:
The review focuses on glycosphingolipids (GSLs), particularly gangliosides, in osteoarthritis (OA) and cartilage regeneration, which is an original contribution since GSLs' role in articular cartilage, OA pathogenesis, and therapeutic prospects has not been extensively covered in existing literature. The unique angle of targeting GSLs for OA treatment and cartilage regeneration therapy represents a novel approach in the field of orthopedic research and regenerative medicine. The paper addresses a significant gap in understanding the specific roles and therapeutic potential of glycosphingolipids, especially gangliosides, in cartilage degeneration and regeneration. It focuses on the enzymatic regulation of ganglioside expression and its implications for cell signaling in chondrocytes and progenitor cells.
2. Addition to Subject Areal:
This review synthesizes and discusses the impact of GSLs on cartilage homeostasis, the role of GSLs in cartilage repair and differentiation, and the potential therapeutic implications of targeting GSLs in OA and cartilage injury treatments. Compared to existing literature, it provides a comprehensive analysis of the molecular mechanisms by which GSLs influence cartilage biology and proposes new therapeutic strategies based on GSL modulation, which is a significant addition to the field.
3. Improvements in methodology:
Regarding the methodology, since this is a review paper it would have been better if a PRISMA was added to state the keywords, number of papers, and rage of years that were looked into
4.Consistency of Conclusions with Evidence and Experiments:
The conclusions are consistent with the evidence presented, highlighting GSLs as crucial components in signaling pathways affecting cartilage metabolism and OA progression. The review systematically addresses the questions posed, particularly through summarizing existing studies on GSL manipulation in animal models and discussing the implications for cartilage regeneration.
5.Appropriateness of References:
The references appear appropriate and relevant, covering a broad range of studies on GSLs, cartilage biology, and OA. They provide a solid foundation for the review's arguments and conclusions.
6.Additional Comments on Tables, Figures:
The tables and figures, such as Figure 1 detailing the biosynthetic pathway for gangliosides and Table 1 summarizing genetic defects in mouse glycan formation, are informative and contribute to the understanding of GSLs' complex role in cartilage biology. The quality of data and how it's presented in the document seem to be high, aiding in the clarity of the review's messages. However, it could also be beneficial if additional figures visualizing the signaling pathways impacted by GSLs or summarizing the outcomes of therapeutic interventions are shown. It might further enhance the reader's comprehension.
Round 2
Reviewer 1 Report
Comments and Suggestions for Authors
Thank you for implementing all the suggested changes.
The title could be more appealing to the reader but it is fine.
Everything is ok.